# Making Your First Choice: To Address Cold Start Problem in Medical Active Learning

**Liangyu Chen**[1]                                                         LIANGYU.CHEN@NTU.EDU.SG
**Yutong Bai**[2]                                                                 YTONGBAI@GMAIL.COM
**Siyu Huang**[3]                                                          HUANG@SEAS.HARVARD.EDU
**Yongyi Lu**[2]                                                                 YYLU1989@GMAIL.COM
**Bihan Wen**[1]                                                               BIHAN.WEN@NTU.EDU.SG
**Alan L. Yuille**[2]                                                       ALAN.L.YUILLE@GMAIL.COM
**Zongwei Zhou**[*2]                                                                ZZHOU82@JH.EDU

[1] *Nanyang Technological University*

[2] *Johns Hopkins University*

[3] *Harvard University*

**Editors:** Accepted for publication at MIDL 2023

## Abstract

Active learning promises to improve annotation efficiency by iteratively selecting the most important data to be annotated first. However, we uncover a striking contradiction to this promise: at the first few choices, active learning fails to select data as efficiently as random selection. We identify this as the cold start problem in active learning, caused by a biased and outlier initial query. This paper seeks to address the cold start problem and develops a novel active querying strategy, named **HaCon**, that can exploit the three advantages of contrastive learning: (1) no annotation is required; (2) label diversity is ensured by pseudo-labels to mitigate bias; (3) typical data is determined by contrastive features to reduce outliers. Experiments on three public medical datasets show that **HaCon** not only significantly outperforms existing active querying strategies but also surpasses random selection by a large margin. Code is available at https://github.com/cliangyu/CSVAL.

## 1. Introduction

Cold start is a crucial problem because if the initial query is not informative enough, the active learning process may require a large number of additional labeled data, resulting in higher annotation costs and slower learning (Yehuda et al., 2022; Lang et al., 2021), as illustrated in Figure 1. However, there is a lack of studies that systematically study the cold start problem, investigate its causes, and provide practical solutions to address it.

Random selection is generally considered a default choice to start active learning because the randomly sampled query is independent and identically distributed (i.i.d.) to the entire data distribution. Maintaining a similar distribution between training and test data is beneficial, particularly when using limited training data (Jadon, 2021). On the other hand, previous active querying strategies cause two problems when selecting initial queries: **(I) Biased query** (the inter-class factor). Active learning tends to select data that is biased toward specific classes. Empirically, Figure 2 reveals that the class distribution in the selected

---

* Corresponding author: Zongwei Zhou

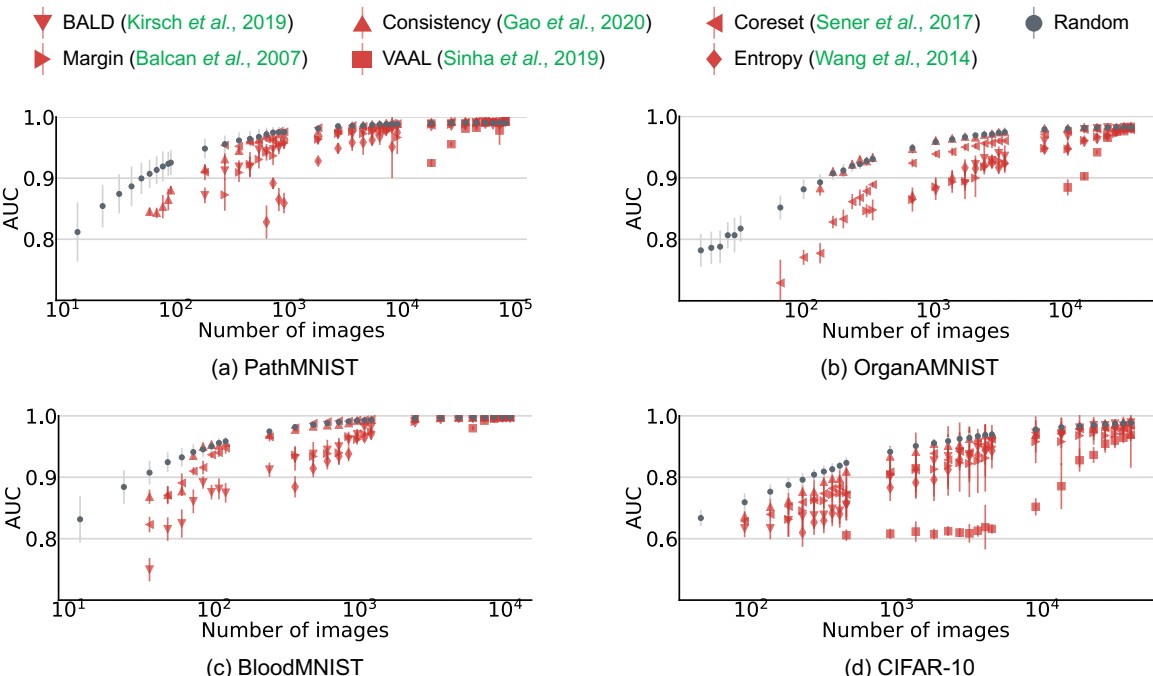

Figure 1: **Cold start problem in active learning.** We found that most active querying strategies are outperformed by random selection in selecting initial queries. Contemporary active learning is typically initialized with a random query and subsequently outperforms random selection by actively seeking the most informative data at each circle. However, if it starts with an actively sampled query associated with biased and outlier data (§1), the model trained with such query may be suboptimal, and therefore consequently the subsequent queries selected by the model will be suboptimal. As a consequence, active learning strategies may continue to underperform random selection until a large amount of labeled data is available.

query is highly unbalanced. These active querying strategies (*e.g.*, Entropy, Margin, VAAL, etc.) can barely outperform random sampling at the beginning because some classes are simply not selected for training. It is because data from the minority classes occur much less frequently than from the majority classes. Moreover, datasets in practice are often highly unbalanced, particularly medical images (Litjens et al., 2017; Zhou et al., 2021a). Such imbalance escalates biased sampling. We hypothesize that the *label diversity* (Figure 2) of a query is an important criterion to determine the importance of the annotation. To pursue label diversity, we exploit the pseudo-labels generated by $K$-means clustering. **(II) Outlier query** (the intra-class factor): Many active querying strategies rely heavily on a trained classifier to produce predictions or features. However, there is no such classifier at the start of active learning, at which point no labeled data is available for training. To utilize the unlabeled data, we consider contrastive learning, which encourages models to discriminate between data augmented from the same image and data from different images (Chen et al., 2020b,a). We hypothesize that contrastive learning can act as an alternative to select typical data and eliminate outliers. Data that is hard to contrast from others could be considered

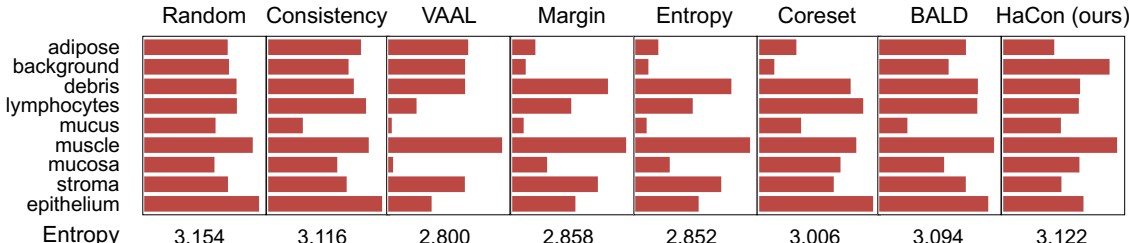

Figure 2: **Label diversity.** Random selection, the leftmost strategy, can reflect the approximate class distribution of the entire dataset. As seen, even with a relatively larger initial query budget (40,498 images, 45% of the dataset), most active querying strategies are biased towards certain classes in the PathMNIST dataset. On the contrary, our **HaCon** selects more data from minority classes while retaining the class distribution of major classes. We compute the entropy value (different from the Entropy-based active selection strategy in Col. 5) to quantify the diversity of the selected classes.

*typical data.* Inspired by Dataset Maps (Swayamdipta et al., 2020), we propose a novel active querying strategy that can effectively select typical data and reduce outliers.

Systematic experiments in §3 confirm that the level of label diversity and the inclusion of typical data are two explicit criteria for determining the annotation importance. Naturally, contrastive learning can approximate these two criteria simultaneously: pseudo-labels in clustering implicitly enforce label diversity in the query; instance discrimination determines typical data. To the best of our knowledge, we are among the first to demonstrate and address the cold start problem in the field of medical image analysis (and perhaps, computer vision), making three contributions: (1) illustrating the cold start problem in active learning, (2) investigating the underlying causes with rigorous empirical analysis and visualization, and (3) determining effective initial queries for the active learning procedure. In addition, compared with random selection, **HaCon** reduces the annotation efforts at the early stage of the active learning procedure by 61%, 21%, and 52% for colon pathology, abdominal CT, and blood cell microscopy images, respectively.

**Related work.** In natural language processing, Yuan et al. (2020) addressed the cold start problem by pre-training models using self-supervision. They attributed the cold start problem to model instability and data scarcity. In vision tasks, although active learning has shown higher performance than random selection (Zhou et al., 2017; Sourati et al., 2019; Gao et al., 2020; Agarwal et al., 2020; Shui et al., 2020; Mayer and Timofte, 2020; Zhou et al., 2021b), there is limited study discussing how to select the initial query when facing the entire unlabeled dataset. A few recent studies indicated the existence of the cold start problem (Lang et al., 2021; Pourahmadi et al., 2021). A series of studies (Hacohen et al., 2022; Yehuda et al., 2022; Sorscher et al., 2022; Nath et al., 2022) continued to propose new strategies for selecting the initial query from the entire unlabeled data and highlighted that typical data (defined in varying ways) can significantly improve the learning efficiency of active learning at a low budget. A concurrent work of ours (Nath et al., 2022) studied warm start active learning for medical image segmentation, proposing a proxy task to rank images.

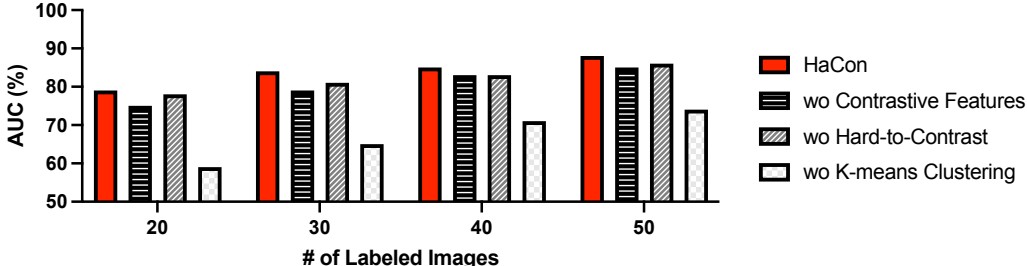

Figure 3: **Ablation of HaCon components.** We ablated three important components in HaCon and compared their performance with **HaCon**. Specifically, we evaluated *wo Contrastive Features*—selecting the samples closest to the centroid K-means on raw data without contrastive learning, *wo Hard-to-Contrast*—selecting the samples closest to the centroid in K-means with contrastive learning rather than the hard-to-contrast data, and *wo K-means clustering*—selecting the top hard-to-contrast samples in the datasets without K-means clustering.

## 2. HaCon

Our proposed active querying strategy **HaCon** consists of three components: (i) extracting features by contrastive learning, (ii) assigning clusters by $K$-means algorithm for label diversity, and (iii) selecting hard-to-contrast data from dataset maps. The ablation study in Figure 3 underlines the significance of each component. We will now introduce each one of the components.

### 2.1. Inter-class Factor: Enforcing Label Diversity to Reduce Biased Query

$K$-**means clustering.** The selected query should cover data from diverse classes, and ideally, select a similar number of data from each class. However, this requires the availability of ground truth, which are inaccessible in active learning. Therefore, we exploit pseudo-labels generated by a simple $K$-means clustering algorithm and select an equal number of data from each cluster to form the initial query to facilitate label diversity. Without knowledge about the exact number of ground-truth classes, over-clustering is suggested in recent works (Van Gansbeke et al., 2020; Zheltonozhskii et al., 2020) to increase performances on datasets with higher intra-class variance. In principle, $K$ (the number of clusters) ought to be larger than the number of classes, and approximates the number of the initial query. The optimal value of K will vary for different datasets. Given 9, 11, and 8 classes in the ground truth, we set $K$ to 30 in our experiments for simplicity. We found that the performance is not sensitive to the choice of $K$ to some extent (detailed ablation study is in Table 17.

**Contrastive features.** Performing $K$-means clustering requires features of each data point. Li et al. suggested that for clustering, contrastive methods (*e.g.*, MoCo, SimCLR, BYOL) are more suitable than generative methods (*e.g.*, reconstruction) because the contrastive feature matrix can be naturally regarded as cluster representations. Therefore, we use MoCo v2 (Chen et al., 2020b)—a popular self-supervised contrastive method—to extract image features. $K$-means and MoCo v2 are certainly not the only choices for clustering and feature

extraction. Figures 2,8,11) show that our querying strategy can yield better label diversity than the other six dominant active querying strategies.

## 2.2. Intra-class Factor: Selecting Typical Data to Avoid Outlier Query

**Dataset map.** Given $K$ clusters generated from §2.1, we now determine which data points ought to be selected from each cluster. Intuitively, a data point can better represent a cluster distribution if it is harder to contrast itself with other data points in this cluster—we consider them typical data. To find these typical data, we modify the original Dataset Map[1] by replacing the ground truth term with a pseudo-label term. For a visual comparison, Figure 4b–c present the difference of Data Maps plotted by ground truths and pseudo-labels. Formally, the modified Data Map can be formulated as follows. Let $\mathcal{D} = \{\boldsymbol{x}_m\}_{m=1}^M$ denote a dataset of $M$ unlabeled images. Considering a minibatch of $N$ images, for each image $\boldsymbol{x}_n$, its two augmented views form a positive pair, denoted as $\tilde{\boldsymbol{x}}_i$ and $\tilde{\boldsymbol{x}}_j$. The contrastive prediction task on pairs of augmented images derived from the minibatch generates $2N$ images, in which a true label $y_n^*$ for an anchor augmentation is associated with its counterpart of the positive pair. We treat the other $2(N-1)$ augmented images within a minibatch as negative pairs. We define the probability of positive pair in the instance discrimination task as:

$$p_{i,j} = \frac{\exp(\mathrm{sim}(\boldsymbol{z}_i, \boldsymbol{z}_j))/\tau}{\sum_{n=1}^{2N} \mathbb{1}_{n \neq i} \exp(\mathrm{sim}(\boldsymbol{z}_i, \boldsymbol{z}_n))/\tau}, \tag{1}$$

$$p_{\theta^{(e)}}(y_n^*|x_n) = \frac{1}{2}[p_{2n-1,2n} + p_{2n,2n-1}], \tag{2}$$

where $\mathrm{sim}(\boldsymbol{u}, \boldsymbol{v}) = \boldsymbol{u}^\top \boldsymbol{v}/\|\boldsymbol{u}\|\|\boldsymbol{v}\|$ is the cosine similarity between $\boldsymbol{u}$ and $\boldsymbol{v}$; $\boldsymbol{z}_{2n-1}$ and $\boldsymbol{z}_{2n}$ denote the projection head output of a positive pair for the input $\boldsymbol{x}_n$ in a batch. $\mathbb{1}_{n \neq i} \in \{0, 1\}$ is an indicator function evaluating to 1 iff $n \neq i$. $\tau$ denotes a temperature parameter. Model is parameterized by $\theta^{(e)}$ at the end of the $e^{\mathrm{th}}$ epoch. We define confidence ($\hat{\mu}_m$) across $E$ epochs as:

$$\hat{\mu}_m = \frac{1}{E} \sum_{e=1}^E p_{\theta^{(e)}}(y_m^*|x_m). \tag{3}$$

The confidence ($\hat{\mu}_m$) is the Y-axis of the Dataset Maps (see Figure 4b-c).

**Hard-to-contrast data.** We consider the data with a low confidence value (Equation 3) as "hard-to-contrast" because they are seldom predicted correctly in the instance discrimination task. Apparently, if the model cannot distinguish a data point from others, this data point is expected to carry typical characteristics that are shared across the dataset (Robinson et al., 2020). Visually, hard-to-contrast data gather in the bottom region of the Dataset Maps and "easy-to-contrast" data gather in the top region. As expected, hard-to-learn data are more typical, possessing the most common visual patterns as the entire dataset, whereas easy-to-learn data appear like outliers (Yehuda et al., 2022; Karamcheti et al., 2021), which may not

---

1. Dataset Map (Chang et al., 2017) analyzes datasets by two measures: *confidence* and *variability*, defined as the mean and standard deviation of the model probability of ground truth along the learning trajectory. Note that the previous studies analyzed existing active querying strategies by using the ground truth. As a result, the values of *confidence* and *variability* in the Dataset Maps could not be computed under the practical active learning setting because the ground truth is a priori unknown. Our modified strategy, however, does not require the availability of ground truth.

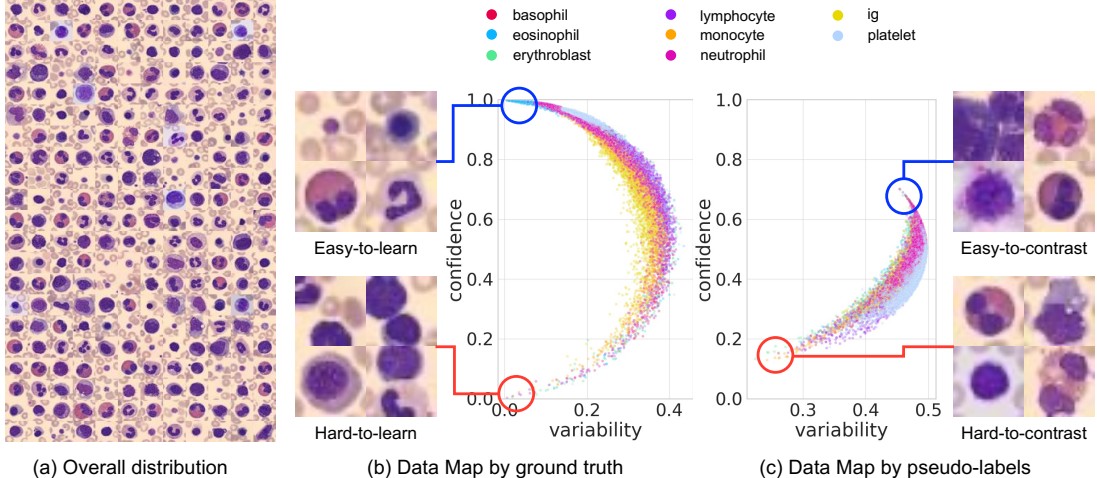

Figure 4: **Dataset Maps.** (a) Dataset overview. (b) Easy- and hard-to-learn data can be selected from the maps based on ground truths. This querying strategy has two limitations: it requires manual annotations and the data are stratified by classes, leading to poor label diversity in the selected queries. (c) Easy- and hard-to-contrast data can be selected from the maps based on pseudo-labels. This querying strategy is label-free and the selected hard-to-contrast data represent the most common patterns in the entire dataset, as presented in (a), therefore are more suitable for training.

follow the majority data distribution (examples in Figure 4a and 4c). Additionally, we also plot the original Dataset Map (Chang et al., 2017; Swayamdipta et al., 2020) in Figure 4b, which grouped data into hard-to-learn and easy-to-learn[2]. Although the results in §3.2 show equally compelling performance achieved by both easy-to-learn (Swayamdipta et al., 2020) and hard-to-contrast data (ours), the latter does not require any manual annotation, and therefore are more practical and suitable for active learning. In practice, we select the same number of top hard-to-contrast data from each cluster.

**Ablation studies.** To study how each component contributes to the performance of HaCon, we exhibit the ablation studies in Figure 3. Diverse sampling from clustering representative embeddings is essential. Meanwhile, Hard-to-Contrast is a useful criterion to improve initial query selection.

## 3. Experimental Results

**Datasets & metrics.** HAM10000 (Tschandl et al., 2018), CIFAR-10-LT (Cui et al., 2019), and three datasets of different modalities in MedMNIST (Yang et al., 2021) are used: PathMNIST, BloodMNIST, OrganAMNIST. The HAM10000 dataset provides dermatoscopic images sized 512×512; the CIFAR-10-LT and MedMNIST datasets offer images sized 28×28

---

2. Swayamdipta et al. (2020) indicated that easy-to-learn data facilitated model training in the low budget regime because easier data reduced the confusion when the model approaching the rough decision boundary. In essence, the advantage of easy-to-learn data in active learning aligned with the motivation of curriculum learning (Bengio et al., 2009).

Table 1: **Diversity is a significant add-on to most querying strategies.** AUC scores of different querying strategies are compared on three medical imaging datasets. In either low budget (*i.e.* 0.5% or 1% of MedMNIST datasets) or high budget (*i.e.*, 10% or 20% of CIFAR-10-LT) regimes, both random and active querying strategies benefit from enforcing the label diversity of the selected data. The cells are highlighted in blue when adding diversity performs no worse than the original querying strategies. Coreset (Sener and Savarese, 2017) works very well as its original form because this querying strategy has implicitly considered the label diversity (also verified in Table 6) by formulating a $K$-center problem, which selects $K$ data points to represent the entire dataset. Some results are missing (marked as "-") because the querying strategy fails to sample at least one data point for each class. Results of more sampling ratios are presented in Appendix Figures 7, 10.

| | | PathMNIST | | OrganAMNIST | | BloodMNIST | |
|---|---|---|---|---|---|---|---|
| | | 0.5% | 1% | 0.5% | 1% | 0.5% | 1% |
| | Unif. | (499) | (899) | (172) | (345) | (59) | (119) |
| Random | ✓ | 96.8±0.6 | 97.6±0.6 | 91.1±0.9 | 93.3±0.4 | 94.7±0.7 | 96.5±0.4 |
| | ✗ | 96.4±1.3 | 97.6±0.9 | 90.7±1.1 | 93.1±0.7 | 93.2±1.5 | 95.8±0.7 |
| Consistency | ✓ | 96.4±0.1 | 97.9±0.1 | 92.3±0.5 | 92.8±1.0 | 92.9±0.9 | 95.9±0.5 |
| | ✗ | 96.2±0.0 | 97.6±0.0 | 91.0±0.3 | 94.0±0.6 | 87.9±0.2 | 95.5±0.5 |
| Margin | ✓ | 97.9±0.2 | 96.0±0.4 | 81.8±1.2 | 85.8±1.4 | 89.7±1.9 | 94.7±0.7 |
| | ✗ | 91.0±2.3 | 96.0±0.3 | - | 85.9±0.7 | - | - |
| Entropy | ✓ | 93.2±1.6 | 95.2±0.2 | 79.1±2.3 | 86.7±0.8 | 85.9±0.5 | 91.8±1.0 |
| | ✗ | - | 87.5±0.1 | - | - | - | - |
| Coreset | ✓ | 95.0±2.2 | 94.8±2.5 | 85.6±0.4 | 89.9±0.5 | 88.5±0.6 | 94.1±1.1 |
| | ✗ | 95.6±0.7 | 97.5±0.2 | 83.8±0.6 | 88.5±0.4 | 87.3±1.6 | 94.0±1.2 |
| BALD | ✓ | 95.8±0.2 | 97.0±0.1 | 87.2±0.3 | 89.2±0.3 | 89.9±0.8 | 92.7±0.7 |
| | ✗ | 92.0±2.3 | 95.3±1.0 | - | - | 83.3±2.2 | 93.5±1.3 |

for proof-of-concept experiments. Area Under the ROC Curve (AUC) and Accuracy are used as the evaluation metrics. All results were based on at least three independent runs, particularly, 100 independent runs for random selection.

**Baselines & implementations.** We benchmark a total of seven querying strategies: (1) random selection, (2) Max-Entropy (Wang and Shang, 2014), (3) Margin (Balcan et al., 2007), (4) Consistency (Gao et al., 2020), (5) BALD (Kirsch et al., 2019), (6) VAAL (Sinha et al., 2019), and (7) Coreset (Sener and Savarese, 2017). For contrastive learning, we trained 200 epochs with MoCo v2, following its default hyperparameter settings. We set $\tau$ to 0.05 in equation 2. To reproduce the large batch size and iteration numbers in Chen et al. (2020a), we apply repeated augmentation (Hoffer et al., 2020; Tong et al., 2022; Touvron et al., 2021) (detailed in Table 4). More implementation details about the dataset split, training recipe, and hyperparameter setting can be found in Appendix A.

## 3.1. Contrastive Features Enforce Label Diversity to Reduce Biased Query

**Label coverage & diversity.** Most active querying strategies have selection bias towards specific classes, thus the class coverage in their selections might be poor (see Table 6), particularly at low budgets. Simply enforcing label diversity to these querying strategies can significantly improve the performance (see Table 1), which suggests that label diversity is one of the causes that existing active querying strategies perform poorer than random selection. Our proposed active querying strategy, however, is capable of covering 100%

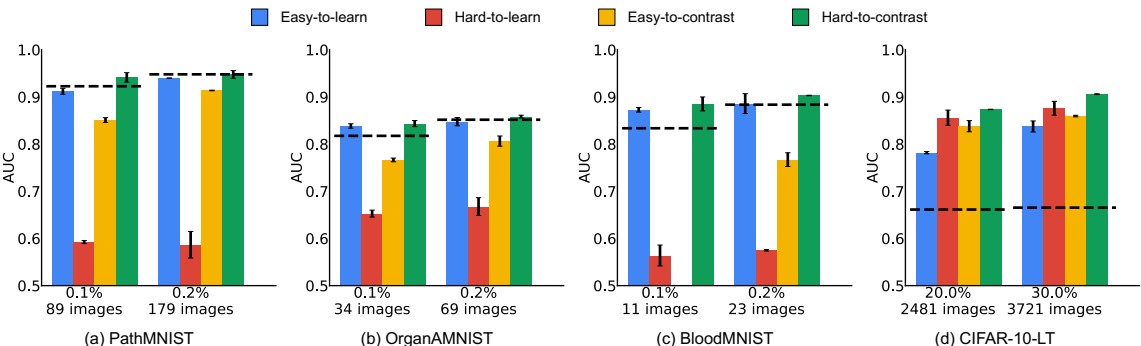

Figure 5: **Quantitative comparison of map-based querying strategies.** Random selection (dot-lines) can be treated as a highly competitive baseline in cold start because it outperforms six contemporary active querying strategies as shown in Figure 1. In comparison with random selection and three other querying strategies, hard-to-contrast performs the best. Although easy-to-learn and hard-to-learn sometimes perform similarly to hard-to-contrast, their selection processes require ground truths (Swayamdipta et al., 2020), which are not available in cold start active learning.

classes in most low budget scenarios ($\leq 0.002\%$ of full dataset) by integrating $K$-means clustering with contrastive features.

### 3.2. Pseudo-labels Select Typical Data to Avoid Outlier Query

**Hard-to-contrast data are practical for the cold start problem.** Figure 5 presents the quantitative comparison of four map-based querying strategies, wherein easy- or hard-to-learn are selected by the maps based on ground truths, easy- or hard-to-contrast are selected by the maps based on pseudo-labels. Note that easy- or hard-to-learn are enforced with label diversity, due to their class-stratified distributions in the projected 2D space (illustrated in Figure 4). Results suggest that *selecting easy-to-learn or hard-to-contrast data contribute to the optimal models.* Either can be regarded as a type of "typical" data that is more informative than atypical data for low-budget active learning (Hacohen et al., 2022). In any case, easy- or hard-to-learn data can not be selected without knowing ground truths, so these querying strategies are not practical for active learning procedures. Selecting hard-to-contrast, on the other hand, is a label-free strategy and yields the highest performance amongst existing active querying strategies (reviewed in Figure 1). More importantly, the hard-to-contrast querying strategy significantly outperforms random selection by 1.8%, 2.6%, and 5.2% on PathMNIST, OrganAMNIST, and BloodMNIST, respectively, by querying 0.1% of the entire dataset. Similarly, on CIFAR-10-LT, hard-to-contrast significantly outperforms random selection by 21.2% and 24.1% by querying 20% and 30% of entire dataset respectively. Note that easy- or hard-to-learn are not enforced with label diversity, for a more informative comparison.

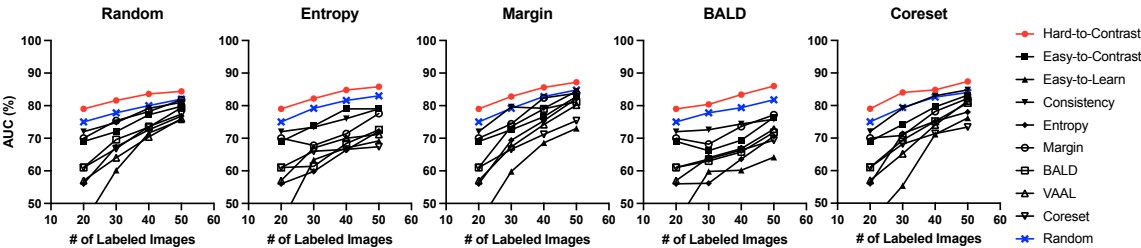

Figure 6: **Active learning performance.** Hard-to-contrast data outperform initial queries, made by either random selection or previous active selection, in every cycle of active learning on OrganAMNIST (as well as other datasets in Figures 14–16). We find that the performance of the initial cycle (20 images) and the last cycle (50 images) are strongly correlated.

### 3.3. On the Importance of Selecting a Superior Initial Query

**A good initial query improves active learning.** We stress the importance of the cold start problem in active learning by conducting correlation analysis. Starting with 20 labeled images as the initial query, the training set is increased by ten more images in each active learning cycle. Figure 6 presents the performance along the active learning (each point in the curve accounts for five independent trials). We finetune all the models based on MoCo v2 pre-trained ResNet-18 checkpoints. The initial query is selected by a total of 9 different strategies[3], and subsequent queries are selected by 5 different strategies. $\text{AUC}_n$ denotes the AUC score achieved by the model that is trained by $n$ labeled images. The Pearson correlation coefficient between $\text{AUC}_{20}$ (starting) and $\text{AUC}_{50}$ (ending) shows a strong positive correlation ($r = 0.92, 0.81, 0.70, 0.82, 0.85$ for random selection, Entropy, Margin, BALD, and Coreset, respectively). This result is statistically significant ($p < 0.05$).

## 4. Conclusion

This paper systematically examines the causes of the cold start problem in active learning and offers a practical and effective solution to address this problem. Analytical results indicate that (1) the level of label diversity and (2) the inclusion of hard-to-contrast data are two explicit criteria to determine the annotation importance. To this end, we devise a novel active querying strategy, HaCon, that can enforce label diversity and determine hard-to-contrast data. The results of three medical imaging datasets show that our initial query not only significantly outperforms existing active querying strategies but also surpasses random selection by a large margin. This finding is intriguing because we discover that the initial query defines the efficacy and efficiency of the subsequent learning procedure. We foresee our solution to the cold start problem as a simple, yet strong, baseline to sample the initial query for active learning in image classification and more medical imaging applications.

---

3. Hard-to-learn is omitted because it falls behind other proposed methods by a large margin (Figure 5).

## Acknowledgments

This work was supported by the Lustgarten Foundation for Pancreatic Cancer Research and partially by the Patrick J. McGovern Foundation Award. The authors appreciate the effort of the OpenMMLab Team to provide and maintain open-source code for the community. The authors want to thank Mingfei Gao for the discussion of initial query quantity and suggestions on the implementation of consistency-based active learning framework. The authors also want to thank Guy Hacohen, Yuanhan Zhang, Akshay L. Chandra, Jingkang Yang, Hao Cheng, Rongkai Zhang, and Junfei Xiao, for their feedback and constructive suggestions at several stages of the project. Computational resources were provided by Machine Learning and Data Analytics Laboratory, Nanyang Technological University. The authors thank the administrator Sung Kheng Yeo for his technical support.

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

## Appendix A. Implementation Configurations

### A.1. Data Split

PathMNIST with nine categories has 107,180 colorectal cancer tissue histopathological images extracted from Kather et al. (2019), with 89,996/10,004/7,180 images for training/validation/testing. BloodMNIST contains 17,092 microscopic peripheral blood cell images extracted from Acevedo et al. (2020) with eight categories, where 11,959/1,712/3,421 images for training/validation/testing. OrganAMNIST consists of the axial view abdominal CT images based on Bilic et al. (2019), with 34,581/6,491/17,778 images of 11 categories for training/validation/testing. CIFAR-10-LT ($\rho$=100) consists of a subset of CIFAR-10 (Krizhevsky et al., 2009), with 12,406/10,000 images for training/testing.

### A.2. Training Recipe for Contrastive Learning

**Pseudocode for our proposed strategy.** The algorithm 1 provides the pseudocode for our proposed hard-to-contrast initial query strategy, as elaborated in §2.

**Pre-training settings.** Our settings mostly follow Chen et al. (2020b); Chen et al.. Table 7($a$) summarizes our contrastive pre-training settings on MedMNIST, following Chen et al. (2020b). Table 7($a$) shows the corresponding pre-training settings on CIFAR-10-LT, following the official MoCo demo on CIFAR-10 (Chen et al.). The contrastive learning model is pre-trained on 2 NVIDIA RTX3090 GPUs with 24GB of memory each. The total number of model parameters is 55.93 million, among which 27.97 million require gradient backpropagation.

**Dataset augmentation.** We apply the same augmentation as in MoCo v2 (Chen et al., 2020b) on all the images of RGB modalities to reproduce the optimal augmentation pipeline proposed by the authors, including PathMNIST, BloodMNIST, CIFAR-10-LT. Because OrganAMNIST is a grey scale CT image dataset, we apply the augmentation in Azizi et al. (2021) designed for radiological images, replacing random grayscale and Gaussian blur with random rotation Table 3 shows the details of data augmentation.

**Repeated augmentation.** Our MoCo v2 pre-training is so fast in computation that data loading becomes a new bottleneck that dominates running time in our setup. We perform repeated augmentation on MedMNIST datasets at the level of dataset, also to enlarge augmentation space and improve generalization. Hoffer et al. (2020) proposed repeated augmentation in a growing batch mode to improve generalization and convergence speed by reducing variances. This approach provokes a challenge in computing resources. Recent works (Hoffer et al., 2020; Touvron et al., 2021; Berman et al., 2019) proved that fixed batch mode also boosts generalization and optimization by increasing the multiplicity of augmentations as well as parameter updates and decreasing the number of unique samples per batch, which holds the batch size fixed. Because the original contrastive learning works (Chen et al., 2020a,b) were implemented on ImageNet dataset, we attempt to simulate the quantity of ImageNet per epoch to achieve optimal performances. The details are shown in Table 4.

We only applied repeated augmentation on MedMNIST, but not CIFAR-10-LT. This is because we follow all the settings of the official CIFAR-10 demo (Chen et al.) in which repeated augmentation is not employed.

## A.3. Training Recipe for MedMNIST and CIFAR-10

**Benchmark settings.** We evaluate the initial queries by the performance of the model trained on the selected initial query, and present the results in Table 1, 7 and Figure 5. The

---

**Algorithm 1:** Active querying hard-to-contrast data

**input:**

$\mathcal{D} = \{\boldsymbol{x}_m\}_{m=1}^M$ {unlabeled dataset $\mathcal{D}$ contains $M$ images}

annotation budget $B$; the number of clusters $K$; batch size $N$; the number of epochs $E$

constant $\tau$; structure of encoder $f$, projection head $g$; augmentation $\mathcal{T}$

$\theta^{(e)}, e \in [1, E]$ {model parameters at epoch $e$ during contrastive learning}

**output:**

selected query $\mathcal{Q}$

$\mathcal{Q} = \varnothing$

**for** epoch $e \in \{1, \ldots, E\}$ **do**

  **for** sampled minibatch $\{\boldsymbol{x}_n\}_{n=1}^N$ **do**

    **for all** $n \in \{1, \ldots, N\}$ **do**

      draw two augmentation functions $t \sim \mathcal{T}$, $t' \sim \mathcal{T}$

      # the first augmentation

      $\tilde{\boldsymbol{x}}_{2n-1} = t(\boldsymbol{x}_n)$

      $\boldsymbol{h}_{2n-1} = f(\tilde{\boldsymbol{x}}_{2n-1})$          # representation

      $\boldsymbol{z}_{2n-1} = g(\boldsymbol{h}_{2n-1})$          # projection

      # the second augmentation

      $\tilde{\boldsymbol{x}}_{2n} = t'(\boldsymbol{x}_n)$

      $\boldsymbol{h}_{2n} = f(\tilde{\boldsymbol{x}}_{2n})$          # representation

      $\boldsymbol{z}_{2n} = g(\boldsymbol{h}_{2n})$          # projection

    **end for**

    **for all** $i \in \{1, \ldots, 2N\}$ and $j \in \{1, \ldots, 2N\}$ **do**

      $s_{i,j} = \boldsymbol{z}_i^\top \boldsymbol{z}_j / (\|\boldsymbol{z}_i\| \|\boldsymbol{z}_j\|)$      # pairwise similarity

      $p_{i,j} = \frac{\exp(s_{i,j})/\tau}{\sum_{n=1}^{2N} \mathbb{1}_{n \neq i} \exp(s_{i,n})/\tau}$    # predicted probability of contrastive pre-text task

    **end for**

    $p_{\theta^{(e)}}(y_n^* | x_n) = \frac{1}{2}[p_{2n-1,2n} + p_{2n,2n-1}]$

  **end for**

**end for**

**for** unlabeled images $\{\boldsymbol{x}_m\}_{m=1}^M$ **do**

  $\hat{\mu}_m = \frac{1}{E} \sum_{e=1}^E p_{\theta^{(e)}}(y_m^* | x_m)$

  Assign $\boldsymbol{x}_m$ to one of the clusters computed by $K$-mean($\boldsymbol{h}, K$)

**end for**

**for all** $k \in \{1, \ldots, K\}$ **do**

  sort images in the cluster $K$ based on $\hat{\mu}$ in an ascending order

  query labels for top $B/K$ samples, yielding $Q_k$

  $\mathcal{Q} = \mathcal{Q} \cup \mathcal{Q}_k$

**end for**

**return** $\mathcal{Q}$

Table 2: Contrastive learning settings on MedMNIST and CIFAR-10-LT.

| config | value |
|---|---|
| backbone | ResNet-50 |
| optimizer | SGD |
| optimizer momentum | 0.9 |
| weight decay | 1e-4 |
| base learning rate† | 0.03 |
| learning rate schedule | cosine decay |
| warmup epochs | 5 |
| epochs | 200 |
| repeated sampling (Hoffer et al., 2020) | see Table 4 |
| augmentation | see Table 3 |
| batch size | 4096 |
| queue length (Chen et al., 2020b) | 65536 |
| $\tau$ (equation 1) | 0.05 |

(*a*) MedMNIST pre-training

| config | value |
|---|---|
| backbone | ResNet-50 |
| optimizer | SGD |
| optimizer momentum | 0.9 |
| weight decay | 1e-4 |
| base learning rate† | 0.03 |
| learning rate schedule | cosine decay |
| warmup epochs | 5 |
| epochs | 800 |
| repeated sampling (Hoffer et al., 2020) | none |
| augmentation | see Table 3 |
| batch size | 512 |
| queue length (Chen et al., 2020b) | 4096 |
| $\tau$ (equation 1) | 0.05 |

(*b*) CIFAR-10-LT pre-training

† $lr$ = $base_lr$ × batchsize / 256 per the linear $lr$ scaling rule (Goyal et al., 2019).

Table 3: **Data augmentations.**

| augmentation | value |
|---|---|
| hflip | |
| crop | [0.08, 1] |
| color jitter | [0.4, 0.4, 0.4, 0.1], p=0.8 |
| grayscale | |
| Gaussian blur | $\sigma_{min}$=0.1, $\sigma_{max}$=2.0, p=0.5 |

(*c*) Augmentations for RGB images

| augmentation | value |
|---|---|
| hflip | |
| crop | [0.08, 1] |
| color jitter | [0.4, 0.4, 0.4, 0.1], p=0.8 |
| rotation | degrees=45 |

(*d*) Augmentations for OrganAMNIST

benchmark experiments are performed on NVIDIA RTX 1080 GPUs, with the following settings in Table 5.

**Cold start settings for existing active querying criteria.** To compare the cold start performance of active querying criteria with random selection (Figure 1), we trained a model with the test set and applied existing active querying criteria.

Table 4: **Repeated augmentation.** For a faster model convergence, we apply repeated augmentation (Hoffer et al., 2020; Tong et al., 2022; Touvron et al., 2021) on MedMNIST by reproducing the large batch size and iteration numbers.

| | # training | repeated times | # samples per epoch |
|---|---|---|---|
| ImageNet | 1,281,167 | 1 | 1,281,167 |
| PathMNIST | 89,996 | 14 | 1,259,944 |
| OrganAMNIST | 34,581 | 37 | 1,279,497 |
| BloodMNIST | 11,959 | 105 | 1,255,695 |
| CIFAR-10-LT($\rho$=100) | 12,406 | 1 | 12,406 |

Table 5: **Benchmark settings.** We apply the same settings for training MedMNIST, CIFAR-10, and CIFAR-10-LT.

| config | value |
|---|---|
| backbone | Inception-ResNet-v2 |
| optimizer | SGD |
| learning rate | 0.1 |
| learning rate schedule | reduce learning rate on plateau, factor=0.5, patience=8 |
| early stopping patience | 50 |
| max epochs | 10000 |
| augmentation | flip, p=0.5 |
| | rotation, p=0.5, in 90, 180, or 270 degrees |
| | reverse color, p=0.1 |
| | fade color, p=0.1, 80% random noises + 20% original image |
| batch size | 128 |

## Appendix B. Additional Results on MedMNIST

### B.1. Label Diversity is a Significant Add-on to Most Querying Strategies

As we present in Table 1, label diversity is an important underlying criterion in designing active querying criteria. We plot the full results on all three MedMNIST datasets in Figure 7. Most existing active querying strategies became more performant and robust in the presence of label diversity.

### B.2. Contrastive Features Enable Label Diversity to Mitigate Bias

Our proposed active querying strategy is capable of covering the majority of classes in most low budget scenarios by integrating K-means clustering and contrastive features, including the tail classes (*e.g.* femur-left, basophil). Compared to the existing active querying criteria, we achieve the best class coverage of selected query among at all budgets presented in Table 6.

Table 6: **Class coverage of selected data.** Compared with random selection (i.i.d. to entire data distribution), most active querying strategies contain selection bias to specific classes, so the class coverage in their selections might be poor, particularly using low budgets. As seen, using 0.002% or an even smaller proportion of MedMNIST datasets, the class coverage of active querying strategies is much lower than random selection. By integrating $K$-means clustering with contrastive features, our querying strategy is capable of covering 100% classes in most scenarios using low budgets ($\leq$0.002% of MedMNIST). We also found that our querying strategy covers most of the classes in the CIFAR-10-LT dataset, which is deliberately more imbalanced.

| | PathMNIST | | OrganAMNIST | | BloodMNIST | | CIFAR-10-LT | |
|---|---|---|---|---|---|---|---|---|
| | 0.00015% | 0.00030% | 0.001% | 0.002% | 0.001% | 0.002% | 0.2% | 0.3% |
| | (13) | (26) | (34) | (69) | (11) | (23) | (24) | (37) |
| Random | **0.79±0.11** | 0.95±0.07 | 0.91±0.08 | 0.98±0.04 | 0.70±0.13 | 0.94±0.08 | 0.58±0.10 | 0.66±0.12 |
| Consistency | 0.78 | 0.88 | 0.82 | 0.91 | 0.75 | 0.88 | 0.50 | 0.70 |
| VAAL | 0.11 | 0.11 | 0.18 | 0.18 | 0.13 | 0.13 | 0.30 | 0.30 |
| Margin | 0.67 | 0.78 | 0.73 | 0.82 | 0.63 | 0.75 | 0.60 | 0.70 |
| Entropy | 0.33 | 0.33 | 0.45 | 0.73 | 0.63 | 0.63 | 0.40 | 0.70 |
| Coreset | 0.66 | 0.78 | 0.91 | 1.00 | 0.63 | 0.88 | 0.60 | 0.70 |
| BALD | 0.33 | 0.44 | 0.64 | 0.64 | 0.75 | 0.88 | 0.60 | 0.70 |
| Ours | 0.78 | **1.00** | **1.00** | **1.00** | **1.00** | **1.00** | **0.70** | **0.80** |

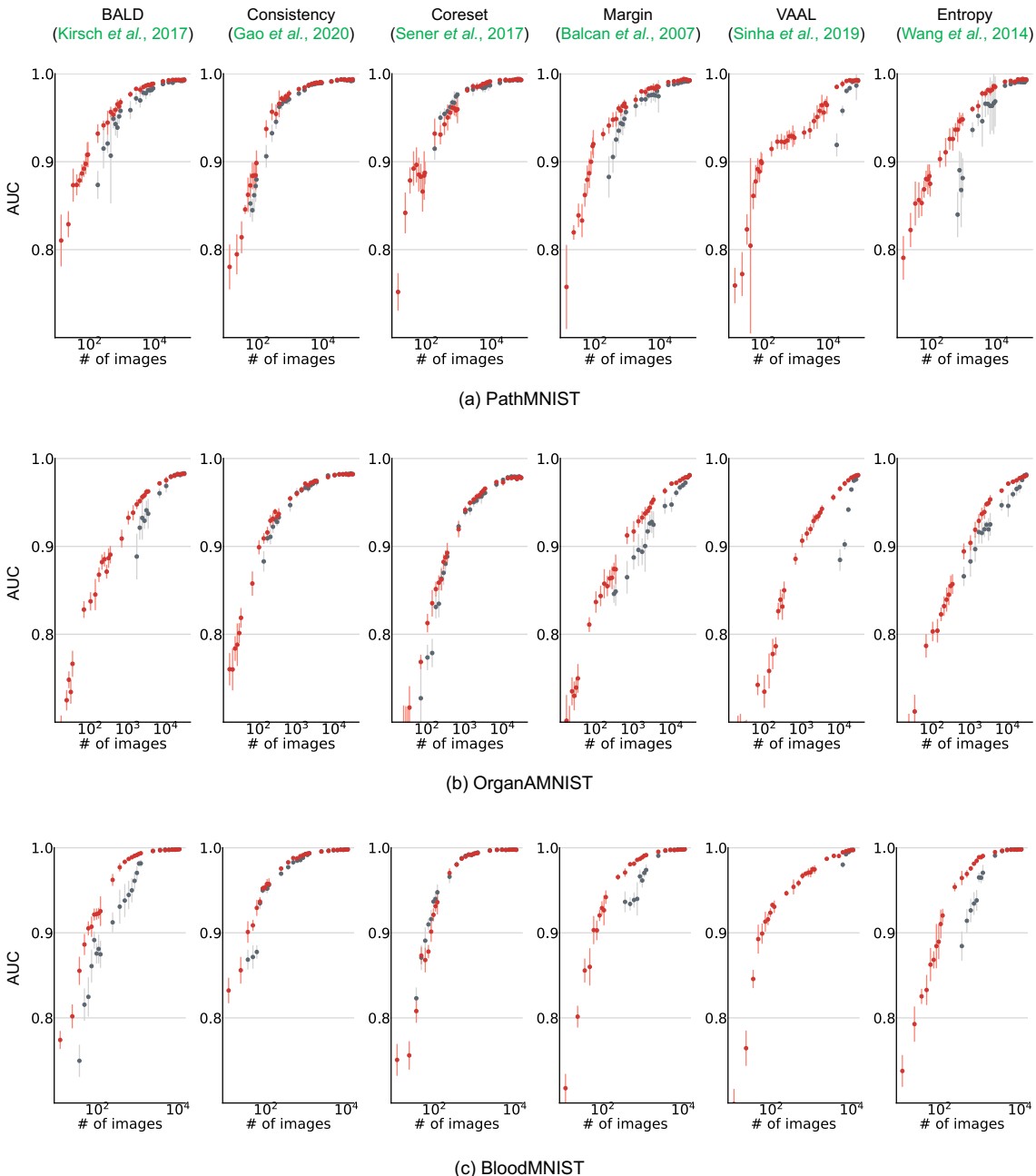

Figure 7: [Extended from Table 1] **Label diversity yields more performant and robust active querying strategies.** The experiments are conducted on three datasets in MedMNIST. The red and gray dots denote AUC scores of different active querying strategies with and without label diversity, respectively. Most existing active querying strategies became more performant and robust in the presence of label diversity, *e.g.* BALD, Margin, VAAL, and Uncertainty in particular. Some gray dots are not plotted in the low budget regime because there are classes absent in the queries due to selection bias.

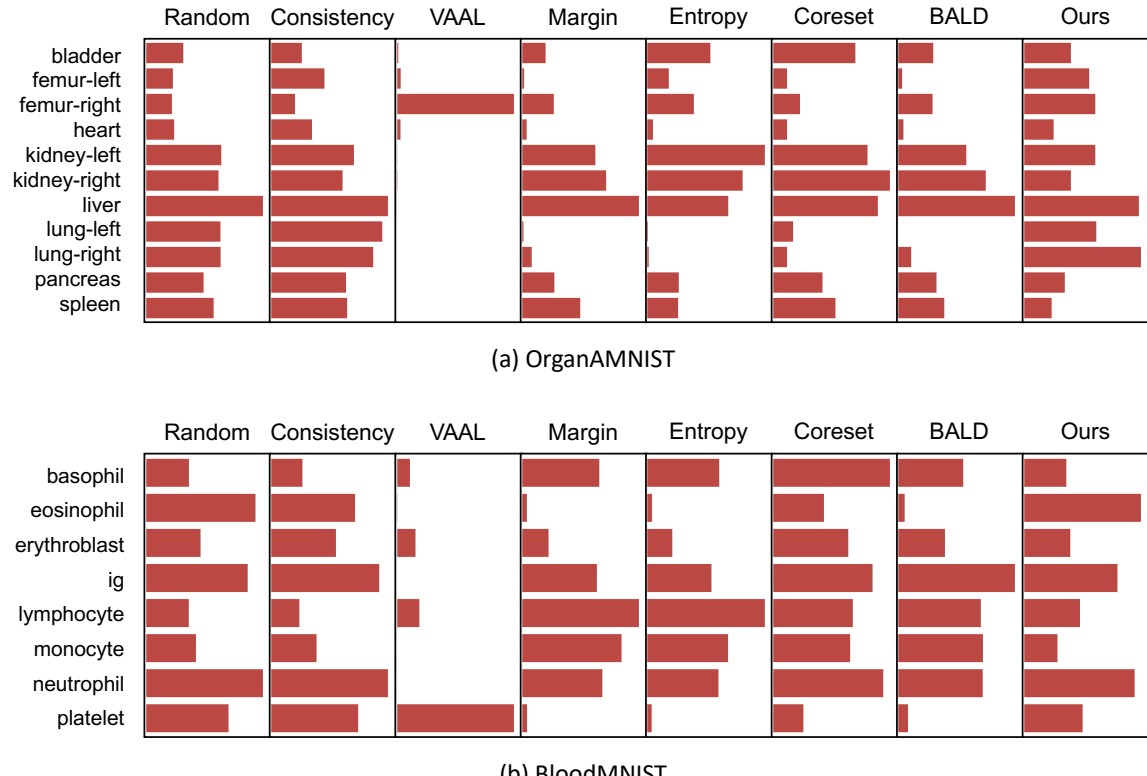

(a) OrganAMNIST

(b) BloodMNIST

Figure 8: [Continued from Figure 2] **Our querying strategy yields better label diversity.** Random on the leftmost denotes the class distribution of randomly queried samples, which can also reflect the approximate class distribution of the entire dataset. As seen, even with a relatively larger initial query budget (691 images, 2% of OrganAMNIST, and 2,391 images, 20% of BloodMNIST), most active querying strategies are biased towards certain classes. For example in OrganAMNIST, VAAL prefers selecting data in the femur-right and platelet classes, but largely ignores data in the lung, liver and monocyte classes. On the contrary, our querying strategy not only selects more data from minority classes (e.g., femur-left and basophil) while retaining the class distribution of major classes.

**Selected query visualization.** To ease the analysis, we project the image features (extracted by a trained MoCo v2 encoder) onto a 2D space by UMAP (McInnes et al., 2018). The assigned pseudo labels have a large overlap with ground truths, suggesting that the features from MoCo v2 are quite discriminative for each class. Overall, Figure 9 shows that hard-to-contrast queries have a greater spread within each cluster than easy-to-contrast ones. Both strategies can cover 100% classes. Nevertheless, we notice that easy-to-contrast selects *local outliers* in clusters: samples that do not belong to the majority class in a cluster. Such behavior will invalidate the purpose of clustering, which is to query uniformly by separating classes. Additionally, it possibly exposes the risk of introducing out-of-distribution data to the query, which undermines active learning (Karamcheti et al., 2021).

**Uncertainty analysis.** The uncertainty of machine learning is divided into two sources: *epistemic uncertainty* that comes from the lack of knowledge by the learner, and *aleatoric uncertainty* that comes from the intrinsic randomness of data (Kendall and Gal, 2017). While epistemic uncertainty is more critical in active learning (Osband et al., 2021), it increases for out-of-distribution outliers (Nguyen et al., 2019). The uncertainty analysis explained why initiating active learning with outliers is not beneficial.

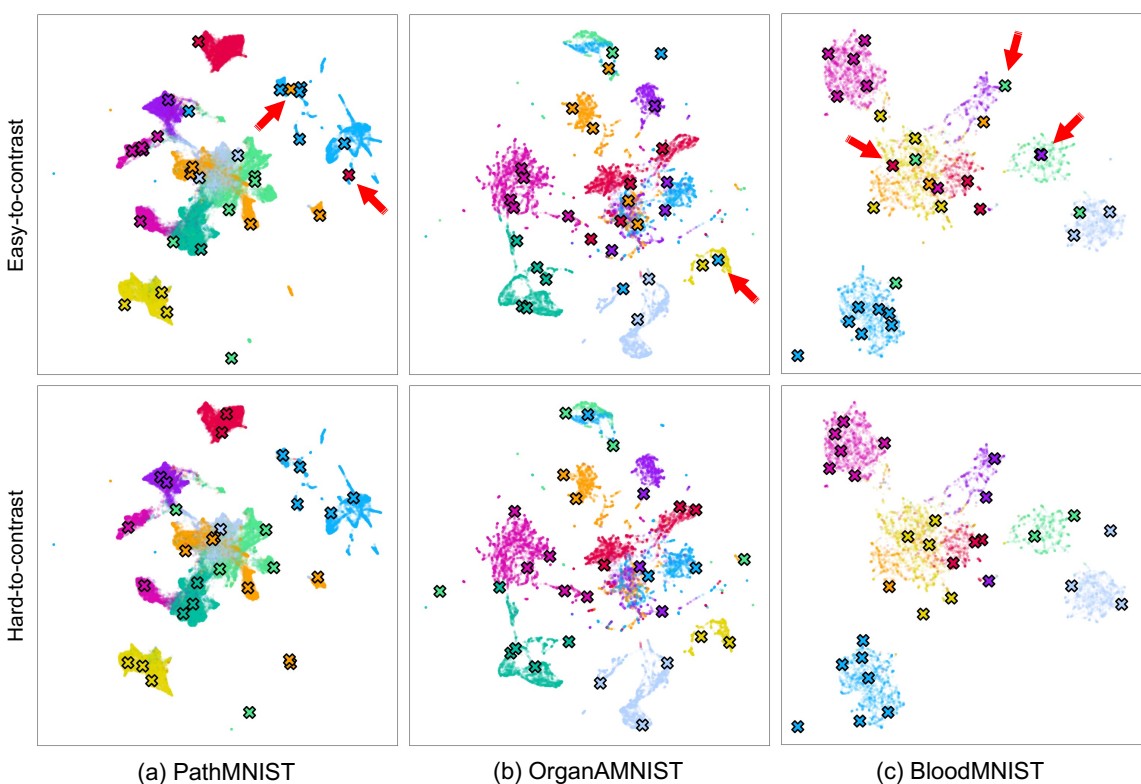

(a) PathMNIST  (b) OrganAMNIST  (c) BloodMNIST

Figure 9: **Visualization of $K$-means clustering and our active selection.** UMAP (McInnes et al., 2018) is used to visualize the feature clustering. Colors indicate the ground truth. Contrastive features clustered by the $K$-means algorithm present a fairly clear separation in the 2D space, which helps enforce the label diversity without the need for ground truth. The crosses denote the selected easy- (top) and hard-to-contrast (bottom) data. Overall, hard-to-contrast data have a greater spread within each cluster than easy-to-contrast ones. In addition, we find that easy-to-contrast tends to select outlier classes that do not belong to the majority class in a cluster (see red arrows). This behavior will invalidate the purpose of clustering and inevitably jeopardize the label diversity.

# Appendix C. Experiments on CIFAR-10 and CIFAR-10-LT

## C.1. Label Diversity is a Significant Add-on to Most Querying Strategies

As illustrated in Table 7 and Figure 10, label diversity is an important underlying criterion in designing active querying criteria on CIFAR-10-LT, an extremely imbalanced dataset. We compare the results of CIFAR-10-LT with MedMNIST datasets Figure 7. CIFAR-10-LT is more imbalanced than MedMNIST, and the performance gain and robustness improvement of label diversity CIFAR-10-LT is significantly larger than MedMNIST. Most of the active querying strategies fail to query all the classes even at relatively larger initial query budgets.

Table 7: **Diversity is a significant add-on to most querying strategies.** AUC scores of different querying strategies are compared on CIFAR-10 and CIFAR-10-LT. In the low budget regime (*e.g.* 10% and 20% of the entire dataset), active querying strategies benefit from enforcing the label diversity of the selected data. The cells are highlighted in blue when adding diversity performs no worse than the original querying strategies. Some results are missing (marked as "-") because the querying strategy fails to sample at least one data point for each class. Results of more sampling ratios are presented in Appendix Figure 10.

| | | CIFAR-10-LT | | | | | |
|---|---|---|---|---|---|---|---|
| | | 1% | 5% | 10% | 20% | 30% | 40% |
| | Unif. | (142) | (710) | (1420) | (2841) | (4261) | (5682) |
| Consistency | ✓ | 78.0±1.2 | 90.0±0.1 | 91.4±1.1 | 93.4±0.2 | 93.2±0.2 | 94.6±0.2 |
| | ✗ | - | - | 67.1±17.1 | 88.6±0.3 | 90.4±0.6 | 90.7±0.2 |
| VAAL | ✓ | 80.9±1.0 | 90.3±0.5 | 92.6±0.2 | 93.7±0.4 | 93.9±0.8 | 94.5±0.2 |
| | ✗ | - | - | - | - | - | 77.3±1.6 |
| Margin | ✓ | 81.2±1.8 | 88.7±0.7 | 91.7±0.9 | 93.2±0.2 | 94.5±0.1 | 94.7±0.4 |
| | ✗ | - | - | 81.9±0.8 | 86.3±0.3 | 87.4±0.2 | 88.1±0.1 |
| Entropy | ✓ | 78.1±1.4 | 89.6±0.5 | 92.0±1.2 | 91.9±1.3 | 94.0±0.6 | 94.0±0.7 |
| | ✗ | - | 79.0±1.2 | 65.6±15.6 | 86.4±0.2 | 88.5±0.2 | 89.5±0.7 |
| Coreset | ✓ | 80.8±1.0 | 89.7±1.3 | 91.5±0.4 | 93.6±0.2 | 93.4±0.7 | 94.8±0.1 |
| | ✗ | - | - | 65.9±15.9 | 86.9±0.1 | 88.2±0.1 | 90.3±0.2 |
| BALD | ✓ | 83.3±0.6 | 90.8±0.3 | 92.8±0.1 | 90.8±2.4 | 94.0±0.8 | 94.7±0.4 |
| | ✗ | - | 76.8±2.3 | 64.9±14.9 | 84.7±0.6 | 88.0±0.5 | 88.9±0.1 |

## C.2. Contrastive Features Enable Label Diversity to Mitigate Bias

Our proposed active querying strategy is capable of covering the majority of classes in most low budget scenarios by integrating K-means clustering and contrastive features, including the tail classes (horse, ship, and truck). Compared to the existing active querying criteria, we achieve the best class coverage of selected query among all budgets presented in Table 6. As depicted in Figure 10, our querying strategy has a more similar distribution to the overall distribution of dataset and successfully covers all the classes, with the highest proportion of minor classes (ship and truck) among random selection and all active querying methods.

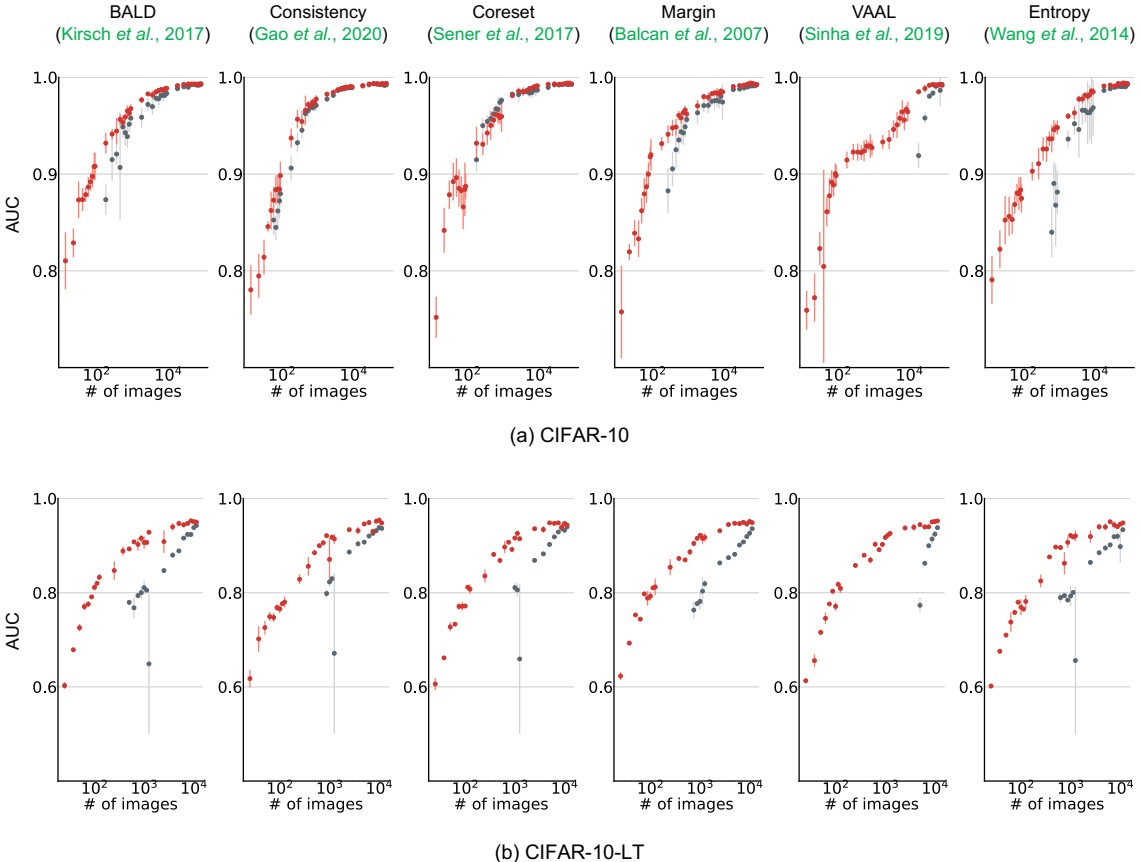

(a) CIFAR-10

(b) CIFAR-10-LT

Figure 10: **Diversity yields more performant and robust active querying strategies.** The experiments are conducted on CIFAR-10-LT. The red and gray dots denote AUC scores of different active querying strategies with and without label diversity, respectively. Observations are consistent with those in medical applications (see Figure 7): Most existing active querying strategies became more performant and robust in the presence of label diversity.

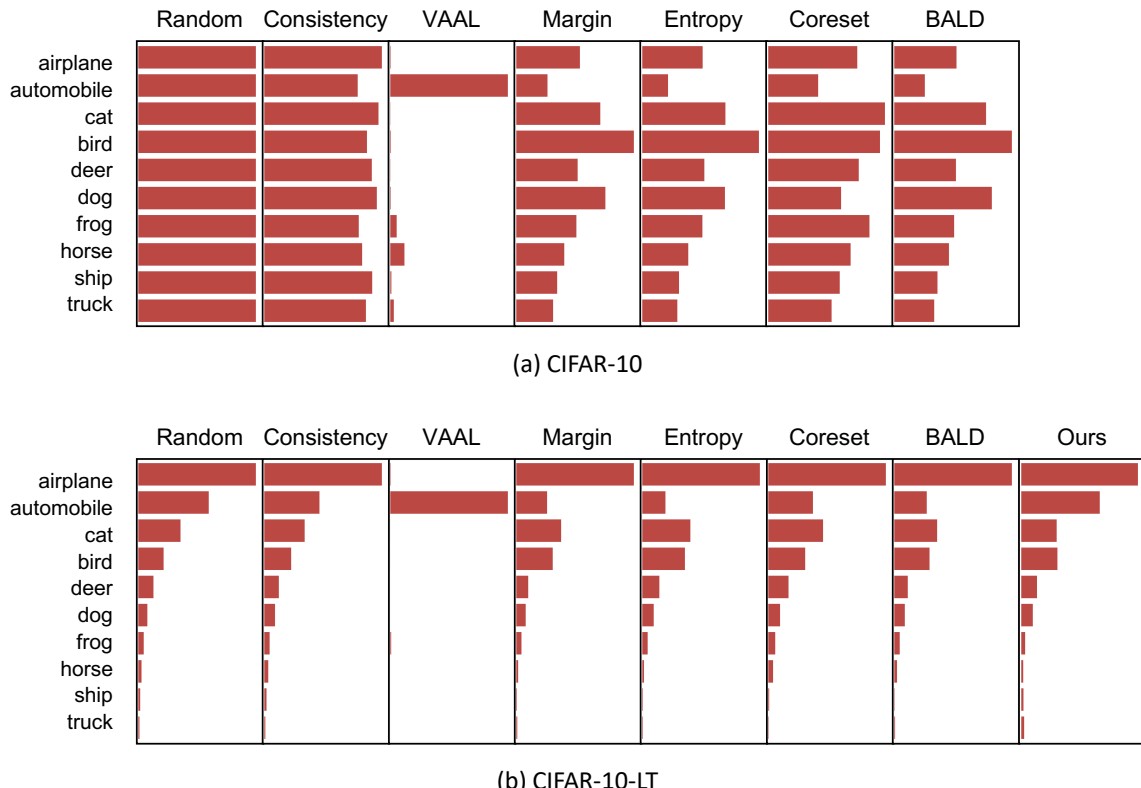

(a) CIFAR-10

(b) CIFAR-10-LT

Figure 11: **Our querying strategy yields better label diversity.** Random on the leftmost denotes the class distribution of randomly queried samples, which can also reflect the approximate class distribution of the entire dataset. As seen, even with a relatively larger initial query budget (5,000 images, 10% of CIFAR-10, and 1420 images, 10% of CIFAR-10-LT), most active querying strategies are biased towards certain classes. Our querying strategy, on the contrary, is capable of selecting more data from the minority classes such as horse, ship, and truck.

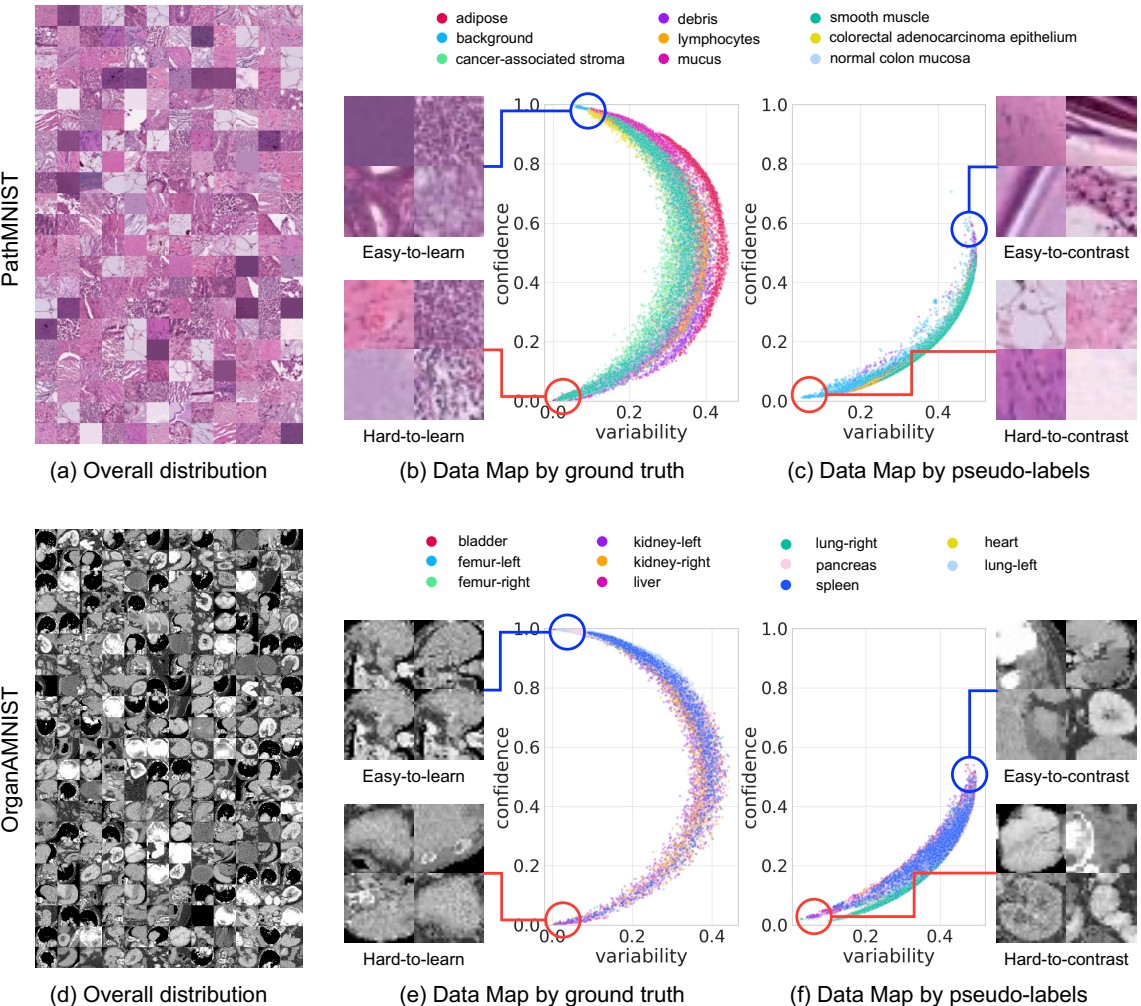

Figure 12: **Active querying based on Dataset Maps.** (a,d) PathMNIST and OrganAMNIST dataset overview. (b,e) Easy- and hard-to-learn data can be selected from the maps based on ground truths (Swayamdipta et al., 2020). This querying strategy has two limitations: (1) requiring manual annotations and (2) data are stratified by classes in the 2D space, leading to poor label diversity in the selected queries. (c,f) Easy- and hard-to-contrast data can be selected from the maps based on pseudo labels. This querying strategy is label-free and the selected "hard-to-contrast" data represent the most common patterns in the entire dataset. These data are more suitable for training and thus alleviate the cold start problem.

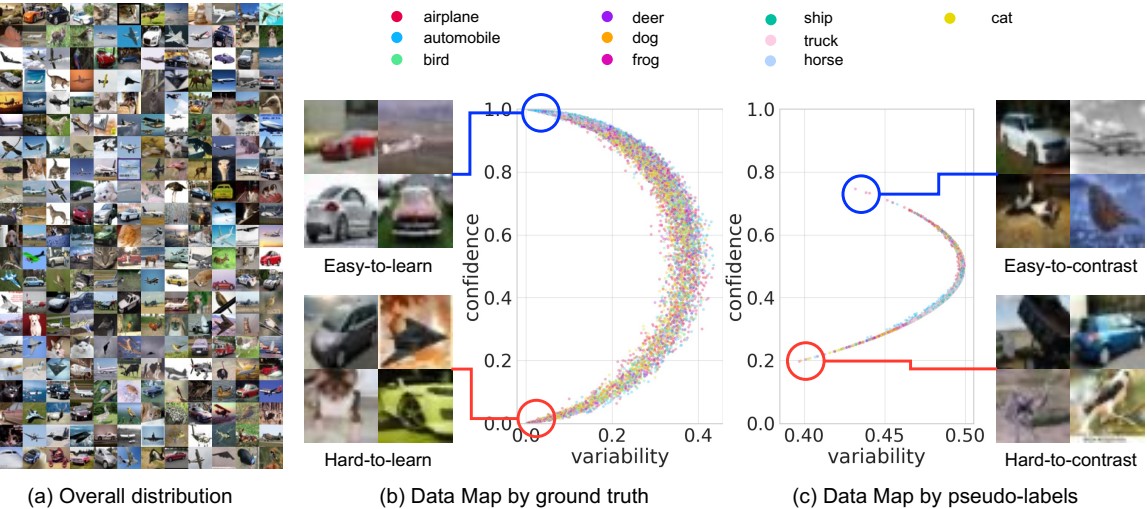

Figure 13: **Active querying based on Dataset Maps.** (a) CIFAR-10-LT dataset overview. (b) Easy- and hard-to-learn data can be selected from the maps based on ground truths (Swayamdipta et al., 2020). This querying strategy has two limitations: (1) requiring manual annotations and (2) data are stratified by classes in the 2D space, leading to poor label diversity in the selected queries. (c) Easy- and hard-to-contrast data can be selected from the maps based on pseudo labels. This querying strategy is label-free and the selected "hard-to-contrast" data represent the most common patterns in the entire dataset. These data are more suitable for training and thus alleviate the cold start problem.

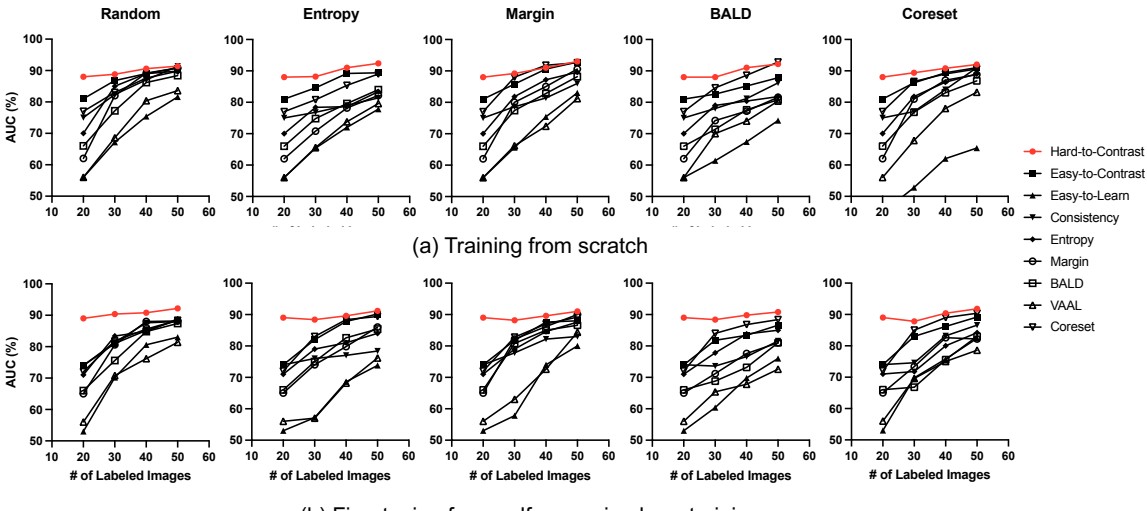

Figure 14: **Performance of each active learning querying strategy with different initial query strategies on BloodMNIST.** Hard-to-contrast initial query strategy (red lines) outperforms other initial query strategies in every cycle of active learning. With each active learning querying strategy, the performance of the initial cycle (20 labeled images) and the last cycle (50 labeled images) are strongly correlated.

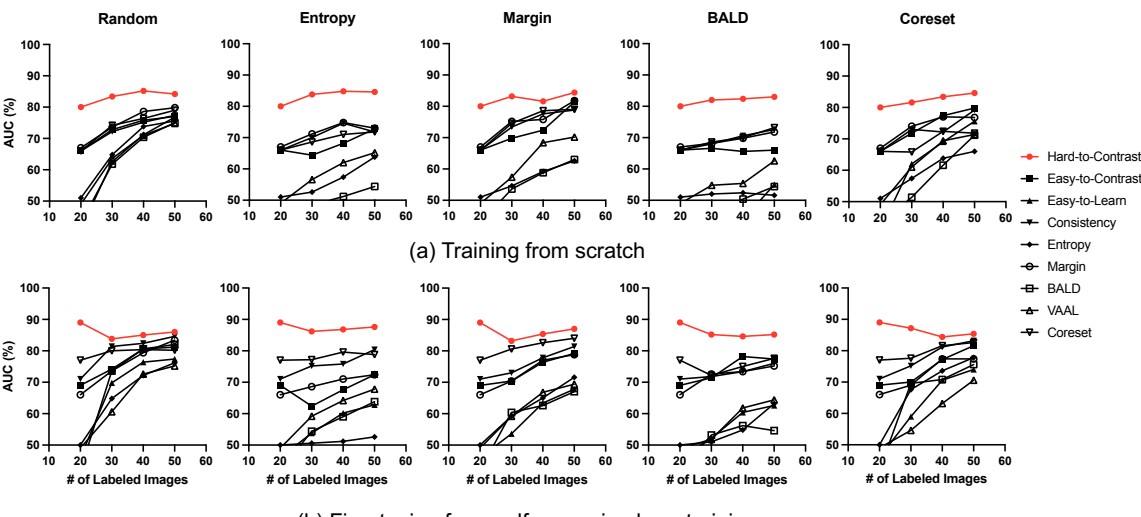

Figure 15: **Performance of each active learning querying strategy with different initial query strategies on PathMNIST.** Hard-to-contrast initial query strategy (red lines) outperforms other initial query strategies in every cycle of active learning. With each active learning querying strategy, the performance of the initial cycle (20 labeled images) and the last cycle (50 labeled images) are strongly correlated.

## Appendix D. Experiments on HAM10000

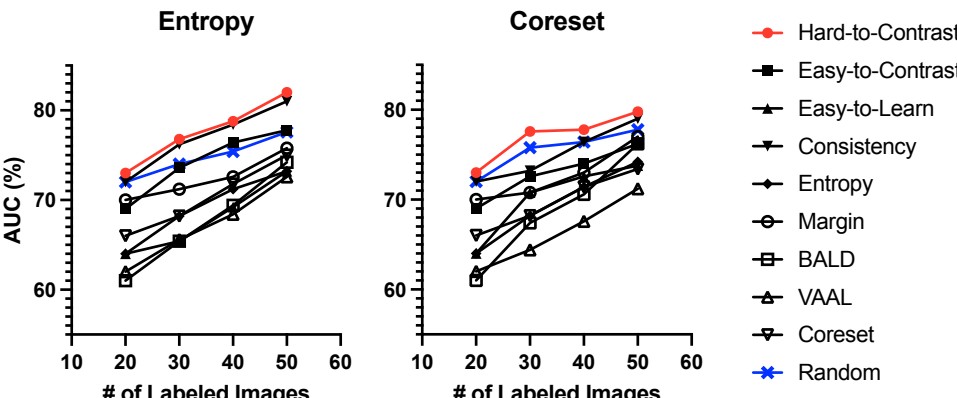

Figure 16: **Performance of different initial query strategies on HAM10000.** Hard-to-contrast initial query strategy (red lines) outperforms other initial query strategies in every cycle of active learning. With each active learning querying strategy, the performance of the initial cycle (20 labeled images) and the last cycle (50 labeled images) are strongly correlated. The active learning querying strategy is Coreset.

## Appendix E. Ablation Studies

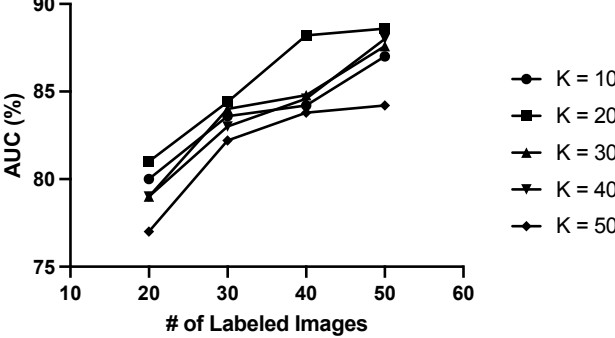

Figure 17: **Ablation of the number of clusters ($K$).** Follows the same settings as Figure 6, we use Coreset to benchmark **HaCon** with different $K$ values. The best performance was achieved when the number of clusters equaled the initial query size (20). To set K equal to the size of initial query is reasonable because it samples diverse data from every cluster. Either under-clustering or over-clustering deteriorates cold start and active learning performances. Overall, the performance was somewhat insensitive to the choice of $K$. The OrganAMNIST dataset was used in this study.

