# OpenReview forum: "Making Your First Choice: To Address Cold Start Problem in Medical Active Learning"
_MIDL.io/2023/Conference — MIDL 2023 Poster_

### Official Review · Reviewer_Eqfg · 2023-02-01

**Confidence:** 5
**Preliminary Rating:** 5
**Recommendation:** Oral

**Summary:**

The author find the problem in active learning which is caused by a biased and outlier initial query, thereby they seek to address the cold start problem and develops a novel active querying strategy. Experiments on three public medical datasets show that the proposed method not only significantly outperforms existing active querying strategies but also surpasses random selection by a large margin.

**Strengths:**

The proposed method can exploit the three advantages of contrastive learning: (1) no annotation is required; (2) label diversity is ensured by pseudo-labels to mitigate bias; (3) typical data is determined by contrastive features to reduce outliers.
The paper is well-writen and easy to follow. The experimental results seems good.

**Weaknesses:**

 Implementation Details: The most concern is the implementation details. The authors have not provided the parameter settings. Did the authors implement a cross-validation procedure to optimize parameter selection for individual baselines? Omission of this step might introduce biases. It would be better if the author can provied more implentation details.

**Deanonymize Review:**

yes

**Paper Type:**

methodological development

**Questions To Address In The Rebuttal:**

see the weakness:
 Implementation Details: The most concern is the implementation details. The authors have not provided the parameter settings. Did the authors implement a cross-validation procedure to optimize parameter selection for individual baselines? Omission of this step might introduce biases. It would be better if the author can provied more implentation details.

---

### Official Review · Reviewer_EAps · 2023-02-03

**Confidence:** 4
**Preliminary Rating:** 2

**Summary:**

This manuscript focuses on solving the cold start problem in active learning, i.e., active learning usually selects data less efficiently than random selection in its first several choices. The authors proposed a querying strategy called HaCon for this problem. The proposed method is compared with random selection and existing active learning methods in three medical datasets.

**Strengths:**

The manuscript addressed the cold start problem in active learning.

The authors proposed a new approach to selecting the initial training set.

The proposed method outperforms random selection and existing active learning methods in the experiments.

**Weaknesses:**

The details about the contribution of the manuscript, the method, and the experimental settings can be better clarified.

An ablation study can be helpful in understanding the contributions of each component of the proposed methods.


**Deanonymize Review:**

no

**Detailed Comments:**

The authors proposed a three-step strategy for selecting the initial query in active learning, i.e., (1) Feature extraction, (2) K-means clustering, and (3) finding hard-to-contrast data. The method looks reasonable. However, it is unclear which steps are this paper's novel contribution. It seems that the initial query with samples in different clusters has been previously proposed. The authors might want to better clarify the contribution.

An ablation study can be helpful in understanding how each component contributes to the performance of the proposed method and why each component is important to the method. The authors might want to compare with (1) Selecting the samples closest to the centroid K-means on raw data without contrastive learning. (2) Selecting the samples closest to the centroid in K-means with contrastive learning rather than the hard-to-contrast data. (3) Selecting the top hard-to-contrast samples in the datasets without K-means.

When finding the hard-to-contrast samples, the method attempts to identify the most Hard-to-contrast sample in each cluster. In the paper, it is not clear that one sample is selected from each cluster.

According to Figure 1, active learning methods underperform the random selection method in some datasets, even if a large number of samples (~$10^4$) are selected. Is it true? Note that CIFAR-10 is a balanced dataset, i.e.,  each class has the same number of samples.

I am confused by Figure 2. It looks like the label diversity of Random selection has better label diversity than the proposed method. Is it true? If this is the case, I doubt whether the proposed method improves label diversity.

In summary, I believe the paper is promising. However, better clarification and additional ablation experiments can further improve the quality of this paper.

**Paper Type:**

methodological development

**Questions To Address In The Rebuttal:**

1. Which components in the method are the novel contributions of this paper?
2. How each component contributes to the performance of the proposed method? why each component is important to the method?
3. Is one sample is selected from each cluster.
4. Does the proposed method get better label diversity than random selection?

---

### Official Review · Reviewer_1grB · 2023-02-04

**Confidence:** 4
**Preliminary Rating:** 4
**Recommendation:** Poster

**Summary:**

The paper discovers that contemporary active learning approaches failed to select data efficiently due to biased initial data selections. The paper then proposes a contrastive learning approach, named HaCon, to initial data selection to ensure that the initial selection of data is diverse and efficient.


**Strengths:**

- The usage of contrastive learning features to construct an unlabeled dataset map for initial data selection is an excellent approach to ensure data diversity.
- The experiments are extensive and show significant improvement compared with other contemporary methods.


**Weaknesses:**

- The image resolution of the benchmark datasets is small (28x28), so they are not representative of the real-world datasets.
- There is no ablation study of how the number of cluster centroids affects performance.
- Lack of random initialization method in Figure 5.


**Deanonymize Review:**

no

**Detailed Comments:**

The reviewer wonders if the author has tried to apply the method to more realistic datasets such as CheXpert, NIH, or MIMIC to see if the proposed method’s performance is still superior to other contemporary approaches.

**Paper Type:**

methodological development

**Questions To Address In The Rebuttal:**

The reviewer suggests the authors add a curve for random initialization to Figure 5 because it is one of the main claims of the paper that other active learning methods’ initial data selection is.
It is not clear how the choice of cluster center affects the method’s performance.

---

### Meta-Review · Area_Chair_hQH9 · 2023-02-20

**Recommendation:** Accept (Poster)
**Confidence:** 5

**Metareview:**

The paper received a weak accept , strong accept and weak reject. The comments raised by Reviewer 2 (weak reject) are valid and warrants greater introspection. However I am not fully convinced by the authors explanation that their contribution is to improve over Datamaps by Swayamdipta. They need to be more explicit about their contributions.  Other reviewers rate the paper highly and I concur  that most of the concerns by R2 and other reviewers have been addressed satisfactorily.